# Renal Tubular CD24 Upregulation Aggravates Folic Acid Induced Acute Kidney Injury: A Possible Role for T Regulatory Cells Inhibition in Mice

**DOI:** 10.3390/jpm13071134

**Published:** 2023-07-13

**Authors:** Moshe Shashar, Doron Schwartz, Asia Zubkov, Sarit Hoffman, Lior Jankelson, Shiran Shapira, Barak Merimsky, Julia Berman, Tamara Chernichovski, Oeren Amitai, Michal Ariela Raz, Rami Hershkovitz, Ayelet Grupper, Talia Weinstein, Nadir Arber, Idit. F. Schwartz

**Affiliations:** 1Departments of Nephrology, Tel Aviv Sourasky Medical Center, 6 Weizmann Street, Tel Aviv 6423906, Israel; shikishashar@gmail.com (M.S.); dorons@tlvmc.gov.il (D.S.); merimsky@gmail.com (B.M.); tom2706@gmail.com (T.C.); orenamitai@gmail.com (O.A.); ayeletg@tlvmc.gov.il (A.G.); taliaw@tlvmc.gov.il (T.W.); nadira@tlvmc.gov.il (N.A.); 2Laniado Hospital, Netanya 4244916, Israel; 3Sackler School of Medicine, Tel Aviv University, Tel Aviv 6139001, Israel; ramih@tlvmc.gov.il; 4Pathology, Tel Aviv Sourasky Medical Center, 6 Weizmann Street, Tel Aviv 6423906, Israel; asiaz@tlvmc.gov.il (A.Z.); sarit.hoffman.levi@gmail.com (S.H.); 5Internal Medicine “T”, Tel Aviv Sourasky Medical Center, 6 Weizmann Street, Tel Aviv 6423906, Israel; lior.yankelson@nyulangone.org (L.J.); julia.berman@gmail.com (J.B.); michalraz123@gmail.com (M.A.R.); 6The Integrated Cancer Prevention Center, Tel Aviv Sourasky Medical Center, 6 Weizmann Street, Tel Aviv 6423906, Israel; shiransha@tlvmc.gov.il

**Keywords:** acute kidney injury, acute tubular necrosis, innate immunity, CD24

## Abstract

Acute kidney injury (AKI) is characterized by cell death and inflammation. CD24 is a protein induced during tissue damage and is not expressed in mature renal tissue. We explored the role of CD24 in the pathogenesis of folic acid-induced AKI (FA-AKI) in mice. A single Intraperitoneal (IP) injection of folic acid induced AKI in WT and CD24^−/−^ mice. Renal function tests, histological analysis, immunohistochemistry, Western blot analysis, and ELISA were performed to assess the severity of renal damage and the intensity of the inflammatory response. FA-AKI induced CD24 in the distal tubular epithelial cells. Compared to WT mice, FA-AKI CD24^−/−^ mice exhibited an attenuated reduction in renal function and histological injury, lower serum IL-10 and interferon γ, and decreased expression of renal TNFα. In contrast, renal and systemic IL-33 upregulation were augmented. CD24^−/−^ FA-AKI animals exhibited increased splenic margination and renal infiltration of regulatory T cells (Tregs). At day 7, FA-AKI CD24^−/−^ mice exhibited increased expression of tubular pro-apoptotic and decreased anti-apoptotic proteins compared to WT animals. Anti-CD24 antibody administration to FA-AKI mice attenuated the decrease in renal function as well as the histological injury. Renal biopsies from patients with ATN stained strongly for CD24 in the distal tubules. In conclusion, during AKI, upregulation of CD24 promotes renal inflammation through inhibition of Treg infiltration and diversion of cell death towards necrosis rather than apoptosis. Neutralization of CD24 may prove a target for future therapies in AKI.

## 1. Introduction

Acute kidney injury (AKI) is manifested by an acute decrease in renal function. Inflammation, parenchymal cell loss, and nephron loss are features of AKI that may eventually lead to tubulointerstitial fibrosis [1]. Mortality is as high as 50% and has changed little over the past three decades. Currently, no satisfactory treatment has been demonstrated to be effective for preventing AKI or reducing the high mortality and progression to CKD [2,3]. The pathogenesis of AKI is characterized by renal tubular cell death, which is followed by tubular dedifferentiation, proliferation, and regeneration. During AKI, a noninfectious inflammatory response is induced by the release of chemokines from sublethally injured cells and damage-associated molecular patterns (DAMPs) from dying cells [4]. DAMPs represent intracellular components released from necrotic cells. DAMPs bind to and activate membrane-bound Toll-like receptors, instigating a host-specific cascade during the early defense phase of innate immunity. This cascade perpetuates the inflammatory response and aggravates tissue injury [4].

CD24 is a small (38–70 kDa) glycosyl phosphoinositol-anchored protein that is expressed on both hematopoietic and non-hematopoietic cells and in many types of cancer cells. Biologically active CD24 has been identified on the cell surface as well as in the nucleus. As a rule, CD24 tends to be expressed at higher levels in progenitor cells and metabolically active cells and to a lesser extent in terminally differentiated cells [5,6]. Diverse immunological functions for CD24 have been reported. On activated B cells, CD24 functions as a T cell co-stimulator for CD4+ T cell clonal expansion. Likewise, CD24 is highly expressed on immature T cells and weakly expressed on peripheral T cells, but it is upregulated in activated T cells [7]. A functional CD24 gene is required for optimal homeostatic proliferation of T cells in a lymphopenic host [8]. Interestingly, CD24 was demonstrated to be associated with a variety of DAMPs, such as high-mobility group box protein-1, heat-shock proteins, and nucleolins [6]. The role of CD24 in modulating tissue injury during sterile inflammation is confusing. In a mouse model of acetaminophen-induced liver necrosis, CD24 negatively regulates the immune response to proteins released by damaged cells, resulting in attenuation of tissue injury [9]. In contrast, CD24 has been shown to aggravate acute liver injury in autoimmune hepatitis by promoting interferon gamma generation by CD4+ T cells [10]. In the kidney, CD24 is expressed on immature cells and disappears after they have reached their final stage of differentiation. CD24 is absent in normal, mature renal tissue [11]. As kidney injury is characterized by an inflammation process, the release of DAMPs, and intense immunological activation, we sought to explore a possible role for CD24 in the pathogenesis of AKI. We utilized the Folic Acid- induced AKI (FA-AKI) model in CD24 knockout and wild-type mice. This experimental model of acute nephrotoxic tubular necrosis recapitulates the major processes seen in human AKI, including renal cell death, inflammation, and renal cell regeneration [12,13].

## 2. Methods

### 2.1. Animals and Reagents

All animal procedures described in this study were conducted in accordance with the guide for the care and use of laboratory animals published by the Israeli Ministry of Health and approved by the institutional and national care and use of laboratory animals’ committees. All standard reagents were obtained from Sigma Chemical Co St. Louis Missouri, St. Louis, MO, USA, unless indicated otherwise.

### 2.2. Animal Model

Our study groups comprised 12–14-week-old wild-type (WT) C57BL/6J male mice (20–25 gr) (Harlan Laboratories, Jerusalem, Israel) and CD24 knockout (CD24^−/−^) mice on a C57/Black background (25–28 gr), which were kindly provided by Prof. Peter Altevogt, German Cancer Research Center, Heidelberg, Germany, and are bred at the animal facility of the Tel Aviv Medical Center, Tel Aviv. These KO mice are genetically tested on a regular basis by PCR analysis of DNA obtained from tail biopsies at the age of 5 weeks, as described previously [14].

Mice were hosted in temperature- and humidity-controlled cages with constant light-dark cycles of 12 h. They were provided with food and water ad libitum. Subsequently, animals were segregated into the following groups (14–16 mice per experimental group).

Group 1: Control, WT vehicle-treated mice.

Group 2: CD24^−/−^, vehicle-treated mice.

Group 3: WT folic acid-treated mice received a single intraperitoneal injection of folic acid (250 mg/Kg in 0.3 mol/L sodium bicarbonate).

Group 4: CD24^−/−^-treated mice, as in group 3.

Experimental procedures were performed for each group at 1, 3, and 7 days following the administration of FA. These time points were chosen to capture the insult and recovery phases of FA-AKI.

In a second set of experiments, we studied the effects of anti-CD24 antibody treatment in WT mice with AKI (FA-AKI). Mice were injected IP with either rat anti mouse M1/69IgG2b mAb 10 mg/Kg BW (BioLegend, San Diego CA, USA) or rat anti-mouse isotype control antibodies at the same dose as adopted in the experimental group before the induction of AKI and 4 days following FA administration, as described previously [15]. Animals were sacrificed on the specified days.

The third set of experiments was performed to elucidate the relative contribution of T regulatory cell infiltration to protection against AKI. Anti-mouse IL-2 receptor (CD25) mAbs (PC61.5 clone, Invitrogen) were administered (i.p. 250 µg) 24 h prior to the administration of folic acid. Serum creatinine was measured at day 7 in both experimental groups.

In all experiments, kidneys were perfused in situ with cold saline before removal. Animals were euthanized using CO_2_.

### 2.3. Kidney Function Tests

Plasma samples were collected at the time of sacrifice. Serum creatinine was measured using a standard picrate method (Cayman Chemicals, Ann Arbor, MN, USA), and BUN levels were measured using standard laboratory techniques. Immunoperoxidase assay of serum neutrophil gelatinase-associated lipocalin (NGAL), a marker of acute renal tubular injury, was determined by the mouse NGAL ELISA kit according to the manufacturer’s protocol (Immunology Consultants Laboratory, Inc., Portland, OR, USA). n = 8 animals per group.

### 2.4. Renal Histology

Following sacrifice, kidneys were bisected, and one half was fixed in 4% formaldehyde, embedded in paraffin, cross-sectioned (4 µm), and stained with Periodic Acid Schiff (PAS) to be examined by light microscopy. The tubular injury was evaluated by a pathologist who was blinded to the nature of the samples. Evidence of cell injury (loss of brush border or vacuolization), cell desquamation, tubular dilation, and signs of regeneration were scored on a semi-quantitative zero-to-three scale, and results from each item were added to yield the tubular injury score, which has a maximal value of 18. n = 6 animals per group.

### 2.5. Immunohistochemistry

Immunohistochemistry was carried out in paraffin-embedded tissue sections (5 µm thick) from the different experimental groups and patients. The samples were incubated with appropriate dilutions of primary antibodies, including rabbit polyclonal anti-mouse and anti-human CD24, rabbit monoclonal anti mouse FoxP3, and rabbit monoclonal anti-mouse CD3, to evaluate regulatory T cell renal trafficking, and then each section was stained with a panel of antibodies using the ImmPress Reagent (Vector Laboratories) or Optiview DAB IHC detection kit (VENTANA), for mice and human samples, respectively. Control sections consisted of staining without primary antibodies and staining with irrelevant primary antibodies (normal rabbit IgG). The scoring of renal Tregs content was determined by the number of infiltrating FoxP3+ cells/CD3+ cells ratio per field. n = 4 animals per group.

### 2.6. Western Blotting

Tubular activated caspase-3, Bcl-xL, IL-33, and TNFα protein expression were determined by immunoblotting as previously described [16]. Following sacrifice, renal tubuli from all experimental groups were isolated using a sieving technique as described [17,18]. Tubuli were homogenized in ice-cold PBS lysis buffer (pH 7.4) containing protease inhibitors (1 mM phenylmethylsulfonyl fluoride, 4.5 μM leupeptin, and 5 μM aprotinin) (ICN Biomedicals Inc.), 0.01% Triton X-100 and 0.1% SDS, then mechanically homogenized and left on ice for 45 min. Homogenates were subsequently centrifuged, and cell lysates were stored in aliquots at −800 °C. The protein content of each sample was determined by the Lowry method. Equal amounts of protein (30 µg) were prepared in sample buffer (2% SDS, 0.01% bromophenol blue, 25% glycerol, 0.0625 M Tris HCL, pH 6.8, 5% mercaptoethanol) and analyzed on a 7.5% SDS-PAGE gel. The gel was transferred onto Hybond ECL nitrocellulose membranes (Amershan Corp.). Following blocking, membranes were incubated with rabbit polyclonal anti-mouse activated caspase-3 antibody, rabbit monoclonal anti-mouse Bcl-xL (Abcam), mouse monoclonal anti-mouse TNF (R&D systems), and rabbit monoclonal anti-mouse IL-33 antibodies (Abcam) for 1 h at room temperature, washed, and incubated with secondary HRP-conjugated goat anti-rabbit antibody in PBS-T for 1 h. Membranes were subsequently washed three times, for 5 min each, in PBS-T. Membranes were then stripped and re-probed with monoclonal anti-β Actin (MP Biomedicals) or anti-GAPDH (Santa Cruz Biotechnology) antibodies as an internal control. The immunoreactive proteins were visualized by enhanced chemiluminescence and scanned using the MicroChemi Imaging system (DNR Bio-Imaging systems). Relative density measurements were conducted using Image J software (S/N 10000128, CAT. NO. 70-25-00). (n = 3 different experiments).

ELISA:

Serum levels of IL-10, INFγ, and IL-33 were measured using specific Enzyme-Linked Immunoabsorbent Assay (ELISA) kits (R&D systems) according to the manufacturer’s instructions. n = 7 animals per group [16].

### 2.7. Statistical Analysis

Data are presented as mean ± SD. A one-way analysis of variance (ANOVA) with post-hoc Tukey’s test was conducted for comparison between groups in order to assess statistical significance. Statistical significance was calculated using the 2-tailed *t* test; a *p* value of less than 0.05 was considered statistically significant. The data were analyzed using GraphPad Prism, version 7.2, software for Windows.

## 3. Results

Renal CD24 during FA-AKI in mice.

First, we evaluated the presence of CD24 in the different experimental groups. CD24 was absent in WT control mice and, as expected, in all CD24^−/−^ experimental groups. Progressive positive staining for CD24 (both cytoplasmatic and nuclear) restricted to the distal tubular epithelial cells was observed in FA-AKI WT mice (Figure 1a). Using western blotting, tubular CD24 content was negative in all CD24^−/−^ experimental groups (data not shown) and in WT control animals. In contrast, it was increased in FA-AKI WT mice throughout the study, most significantly at day 7 (Figure 1b). Next, we investigated the severity of AKI in response to folic acid administration. We found that on days 1 and 3 serum creatinine and BUN levels increased similarly in wild-type and CD24^−/−^ mice. However, on day 7, renal function continued to deteriorate in WT mice, while it either remained unchanged (BUN) or decreased to baseline values (creatinine) in the CD24^−/−^ group. Likewise, FA-AKI WT animals exhibited elevated serum NGAL levels on day 1, which became maximal on day 3 and decreased on day 7. Comparably, in the CD24^−/−^ group, NGAL increased on day 1 as in the WT group, but it started to normalize on day 3, reaching values that were significantly reduced relative to the WT group (Figure 1c). Accordingly, WT and CD24^−/−^ animals manifested similar histologic injury scores one day following folic acid administration. However, it was significantly higher in WT FA-AKI mice on days 3 and 7 compared with the corresponding CD24^−/−^ animals. The most pronounced difference was the patchy peritubular leukocyte infiltration, which was significantly reduced in the CD24^−/−^ group (Figure 1d).

In order to elucidate the mechanism by which CD24 mediates renal injury in FA-AKI, we studied the behavior of several cytokines that have been previously shown to play a role in the pathogenesis of ATN (Figure 2). The serum levels of INFγ and IL-10 were significantly higher in the WT animals on day 3 compared with the CD24^−/−^ animals, while on day 7 the differences were not statistically significant (Figure 2a). Renal expression of TNFα, evaluated by western blotting, progressively increased in both groups, yet it was significantly lower in the CD24^−/−^ animals throughout the study (Figure 2b). IL-33 is an alarmin released by necroptotic cells that can mediate renal injury. In FA-AKI, kidney IL-33 levels have been shown to be elevated. In our current experiments, we found that renal tubular protein expression of IL-33 was significantly increased at all time points in CD24^−/−^ mice compared with WT mice, most notably at day 1 (Figure 2b). This finding was verified by measuring serum IL-33 in the different experimental groups (Figure 2a).

Programed death of tubular epithelial cells is a classic hallmark of AKI. To study the effects of CD24 on tubular cell apoptosis, activated caspase 3 (pro-apoptotic) (Figure 2c) and Bcl-XL (antiapoptotic) (Figure 2d) expressions were evaluated by immunoblotting in the different experimental groups. It was revealed that on day 1 following folic acid, activated caspase 3 was significantly overexpressed in the WT group, while on day 7, its expression was more pronounced in the CD24^−/−^ animals. By contrast, Bcl-XL exhibited a notable increase in the WT mice on day 7 compared to the CD24^−/−^ animals.

Regulatory T cells (Tregs) are a subtype of T lymphocytes that suppress innate immunity. Since IL-33 enhances mobilization and renal recruitment of Tregs, we sought to study renal Tregs infiltration and splenic margination during the study period.). CD24^−/−^ mice demonstrated a significantly higher renal FoxP3/CD3 ratio than the corresponding WT animals only at day 7 following FA administration (Figure 3b). In the spleen, numerable FoxP3-positive cells were identified in the germinal centers in both WT and CD24^−/−^ animals. We were unable to identify a difference between these two groups. However, Tregs mobilization, evidenced by the number of FoxP3-positive cells per field outside the germinal center, was remarkably higher in CD24^−/−^ compared with WT mice (Figure 3a). Moreover, following the administration of anti-IL-2 receptor (CD25) mAbs, the increase in serum creatinine (Δ creatinine) was significantly more pronounced in the CD24^−/−^ group than in the WT group (Figure 3c).

To further investigate the effects of CD24 in FA-AKI, anti-mouse CD24 antibodies were administered to WT mice and compared with anti-mouse isotype control antibodies (Figure 4). Levels of serum creatinine and BUN on day 7 after FA administration were significantly lower in the anti-CD24 antibody-treated group than in the control group (Figure 4a). In addition, anti-CD24 antibodies significantly attenuated renal leukocyte infiltration (Figure 4b). Moreover, IL-33 expression in kidneys from anti-CD24-treated mice was significantly higher than in control isotype-treated animals (Figure 4c).

Finally, we examined the expression of the CD24 protein in three human kidney biopsies (Figure 5): a. A patient without a history of renal disease who underwent complete nephrectomy due to renal cell carcinoma. b. A patient who underwent deceased donor renal transplantation and developed ATN due to prolonged cold ischemic time. c. A patient presented with severe ATN requiring dialysis support due to prolonged dehydration. No staining was observed in the nephrectomized kidney (a), while we disclosed a significant expression of CD24 in those two patients with histology-proven ATN (b and c). The positive staining for CD24 was seen solely in the distal tubular epithelial cells.

## 4. Discussion

This study provides, for the first time to the best of our knowledge, strong evidence that during AKI, CD24, which is not normally expressed in the mature kidney, is upregulated in the distal tubular epithelial cells, and its attenuation ameliorates kidney damage. Multiple mechanisms participate in the evolution of AKI. Given the fact that the differences between WT and CD24^−/−^ experimental groups were evident predominantly in the late phase of the disease (day 7), we hypothesized that CD24 is involved in either inflammation or regeneration processes. Histology analysis revealed that mononuclear infiltration was indeed more intense in WT mice with FA-AKI compared to the corresponding KO animals. Taken together, these data suggest that during AKI, CD24 is upregulated and perpetuates the inflammatory response, thus aggravating renal damage. Therefore, we pursued experiments trying to explore the behavior of various proteins that were previously shown to play a part in inflammation during AKI.

Overall, Serum levels of interleukin-10, interferon γ, and renal expression of TNFα were significantly higher in the FA-AKI WT mice compared to the corresponding CD24^−/−^ animals. Available data suggest that systemic inflammation plays a role in the pathogenesis of AKI. First, tubular cell death initiates an innate immune response. Necrotic cells release intracellular molecules called damage-associated molecular patterns (DAMPs), which activate resident renal inflammatory cells, followed by recruitment and subsequent infiltration with different subsets of leukocytes, causing them to secrete pro-inflammatory cytokines. These responses further guide various types of inflammatory cells to infiltrate the damaged tissue, which leads to additional cell deaths, forming a vicious cycle of cell death and inflammation. Therefore, the extent of inflammation determines the extent of AKI [19,20]. The higher levels of cytokines observed in the FA-AKI WT mice compared with CD24^−/−^ animals suggests that during AKI, the absence of CD24 activation resulted in a diminished inflammatory response.

In contrast, IL-33 levels demonstrated the opposite pattern. In agreement with previous publications, systemic and renal protein expression of IL-33 increased during FA-AKI [13,20]. Interestingly, its levels were significantly higher in the CD24^−/−^ vs. WT FA-AKI mice, reaching a 1:4 ratio during the initial phase of the disease. Similar differences were also observed in AKI animals treated with anti CD24 antibodies. These data imply that CD24 upregulation in AKI attenuates systemic and renal IL-33 expression. Neutralizing CD24 effects by either utilizing KO animals or by treating WT mice with anti-CD24 antibodies led to augmented expression of both systemic and renal IL-33 during FA-AKI.

IL-33 is a cytokine that belongs to the IL-1 family and is acting as an alarmin during tissue injury [21]. Additionally, IL-33 promotes the recruitment and activation of Th2 and IL-33-specific receptor (ST2)-bearing innate lymphoid cells [22]. Regulatory T cells (Tregs) express the IL-33 receptor ST2, which IL-33 promotes [18]. Similar to macrophages, Tregs are involved in both the development of and recovery from AKI. Tregs represent a small subset of CD4+ cells that express the IL-2 receptor α (CD25). In animal models of ischemic AKI, depletion of Tregs by using anti-CD25 antibodies leads to a more severe AKI via an increase in leukocytes, proinflammatory cytokine levels, and tubular damage [23]. Likewise, anti-CD25 antibody treatment impaired tubular recovery, and this effect was thought to be related to a lack of suppression of the innate immune system. Tregs have also been shown to play a protective role in cisplatin-induced AKI [24]. Adoptive transfer of CD4+ and CD25+ Tregs into mice lacking T cells had beneficial effects, including attenuated renal dysfunction, diminished renal macrophage infiltration, and decreased levels of TNF-a and IL1b [25]. We have found that FA-AKI in the CD24^−/−^ mice is characterized by accelerated Tregs recruitment from the spleen and a more intense Tregs infiltration in the renal interstitium when compared with the corresponding WT mice. Moreover, the administration of neutralizing anti-CD25 antibodies to FA-AKI mice resulted in a worsening in renal function on day 7, which was more significant in CD^−/−^ than in WT animals. Taken together, one can conclude that during AKI, CD24 is activated, resulting in inhibition of Tregs migration to the kidneys, thus perpetuating the sterile inflammatory response that characterizes this syndrome.

Another puzzling observation is the augmented apoptotic activity found in the CD24^−/−^-AKI mice, the experimental group that had a more favorable outcome compared with the corresponding WT animals. During AKI, in addition to the well-described tubular epithelial necrosis, there is also an induction of apoptosis, observed mostly in the medullary thick ascending limb epithelium [26]. Depending on the nature of the injury, hemodynamic changes, ongoing metabolic demands, and local oxygen availability, susceptibility to necrotic or apoptotic injury varies along the nephron.

Cells undergoing apoptosis show characteristic features including chromatin aggregation, nuclear and cytoplasmatic condensation, partition of cytoplasm and nucleus to form membrane-bounded apoptotic bodies with preserved organelle morphology, and little associated inflammation. In necrosis, on the other hand, plasma membrane integrity is lost, intracellular organelles and the entire cell swell and rupture, with the release of DAMPs stimulating and amplifying an inflammatory response with monocytic infiltration, the formation of an infarction, and, after repair, a scar [27]. It is conceivable that control mechanisms exist that determine the type of cell death based on the cause and intensity of injury. With reference to ischemia-reperfusion injury in the outer medulla, the relative contributions of necrosis or apoptosis may depend on the severity of ischemia: prolonged ischemia tends to cause necrosis or accidental cell death because of the resultant negation of the cells’ energy and protein levels; shorter periods of ischemia, partial ischemia, or a higher cellular tolerance to ischemic conditions tend to lead to apoptosis [28]. We suggest that, in our experimental model of AKI, CD24 activation induces more necrosis and less apoptosis, which results in a more severe form of renal damage.

While the insult in AKI is restricted to the proximal tubular cells, CD24 activation was observed almost exclusively in the distal tubules. One could argue how these two processes are linked. During AKI, the epithelial cells of the distal tubules have been shown to be relatively resistant to renal damage, perhaps through the formation of various anti-apoptotic and anti-necrotic proteins. It has also been shown that the distal tubular cells have a broad range of key paracrine and autocrine signals by secreting an array of inflammatory, reparative, and survival cytokines. In a neighborly way. These cytokines and growth factors exert their effects on the vulnerable proximal tubular epithelium [28]. Therefore, one can hypothesize that CD24 activation in the distal tubule potentiates an undesired inflammatory response that predominantly affects the proximal tubule.

In conclusion, the current experiments unveil a novel key process, namely CD24 upregulation in the distal tubule, in the pathogenesis of FA-AKI in mice, which adversely affects kidney outcome. CD24 activation is associated with increased renal and systemic inflammation, decreased Tregs recruitment in the kidney, and attenuated apoptotic activity. Neutralization of CD24 significantly attenuated the renal insult and thus may prove to be an effective future treatment for AKI.

## Figures and Tables

**Figure 1 jpm-13-01134-f001:**
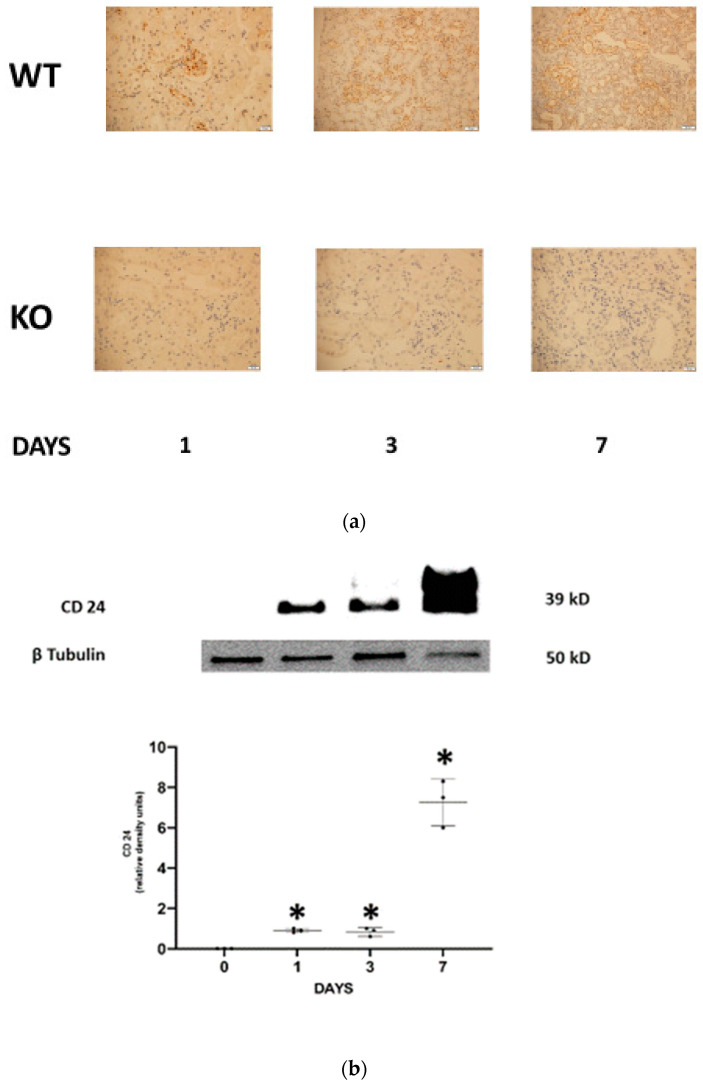
Upregulation of renal tubular CD24 is associated with enhanced renal injury during folic acid-induced AKI. (**a**). Renal CD24 expression was detected by immunohistochemistry in WT and CD24^−/−^ mice with FA-AKI. Progressive positive staining for CD24 restricted predominantly to the distal tubular epithelial cells was observed in the WT FA-AKI mice, while it was absent in the corresponding CD24^−/−^ animals. Four mice per group. Original magnification ×400. Scale bars, 500 µm. (**b**). Representative Western blot and densitometric analysis show regulation of tubular CD24 protein levels following the administration of folic acid in WT mice. Data are presented as the mean ± SD of 3 different experiments * *p* < 0.05 vs. control animals. (**c**). Kidney function following folic acid injury. Renal function was assessed by plasma creatinine, BUN, and NGAL levels. Renal failure was significantly attenuated in the FA-AKI CD24^−/−^ mice in comparison with the corresponding WT animals. The data are presented as the mean ± SD of eight mice per group. * *p* < 0.05 versus the corresponding WT experimental group. (**d**). Decreased histologic kidney injury in CD24^−/−^ mice treated with FA. Representative images of PAS staining and quantification from the different experimental groups. Mean ± SD of six mice per group. Original magnification ×400. Scale bars, 500 µm. * *p* < 0.05 versus the corresponding WT animals. Abbreviations: WT—wild type; KO—knockout. FA—folic acid.

**Figure 2 jpm-13-01134-f002:**
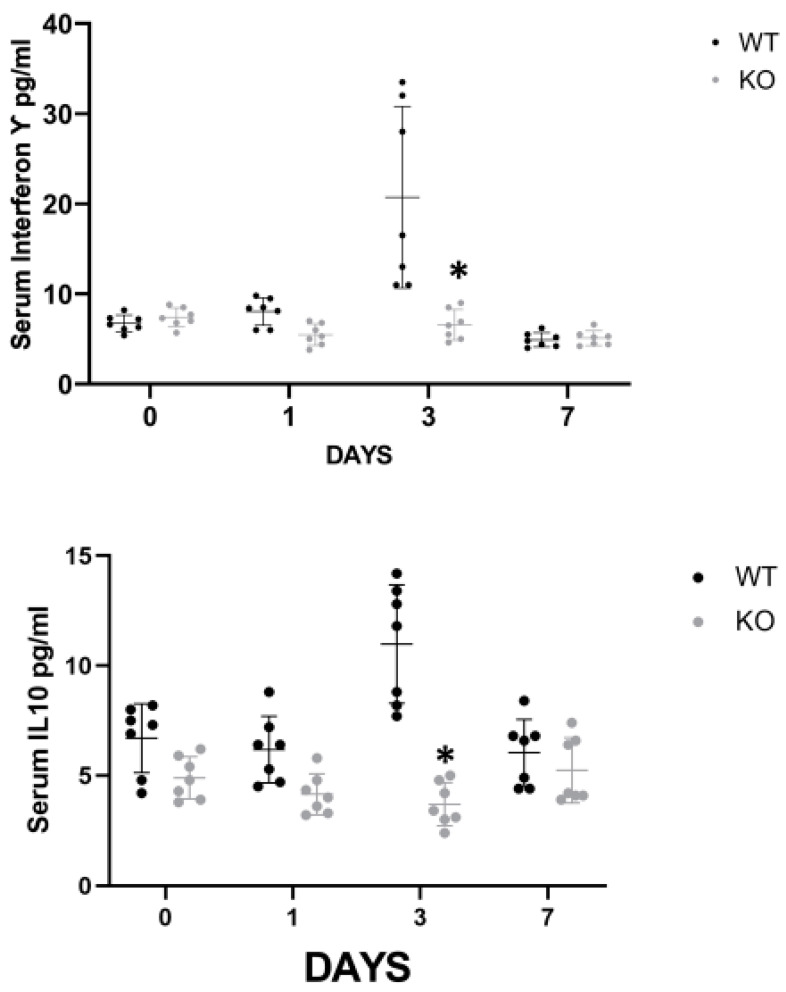
Systemic and renal inflammation are attenuated in FA-AKI CD24^−/−^ mice. (**a**). Dichotomic behavior of serum INF-γ and IL-10 vs. IL-33 during FA-AKI in CD24^−/−^ mice compared with WT animals. The increments in plasma INFγ and IL-10 were attenuated, while IL-33 was increased in CD24^−/−^ mice compared with WT during FA-AKI. Plasma INFγ, IL-10, and IL-33 levels assayed by ELISA. Results are the Mean ± SD of seven animals per group. * *p* < 0.05 vs. the corresponding WT mice. (**b**). Dichotomic behavior of renal tubular protein expression of TNFα and IL-33 during FA-AKI in CD24^−/−^ mice compared with WT animals. The increment in renal tubular TNFα was attenuated while IL-33 increased in CD24^−/−^ mice in comparison with WT mice during FA-AKI. Representative Western blot and densitometric analysis show regulation of TNFα and IL-33 protein levels following the administration of folic acid in CD24^−/−^ and WT mice. The data are presented as the mean ± SD of three different experiments * *p* < 0.05 vs. the corresponding WT animals. Abbreviations: CTL: controls. CD24^−/−^ mice exhibited attenuated early apoptotic activity, which was increased during the late phases of FA-AKI compared to the corresponding WT animals. Representative Western blot and densitometric analysis showing regulation of caspase 3 (pro-apoptotic) (**c**) and Bcl-xL (anti-apoptotic) (**d**) in CD24^−/−^ and WT mice following folic acid administration (FA-AKI). Data are presented as the mean ± SD of 3 different experiments * *p* < 0.05 vs. the corresponding WT animals. Abbreviations: WT—wild type; KO—knockout; FA-AKI—folic acid-induced acute kidney injury.

**Figure 3 jpm-13-01134-f003:**
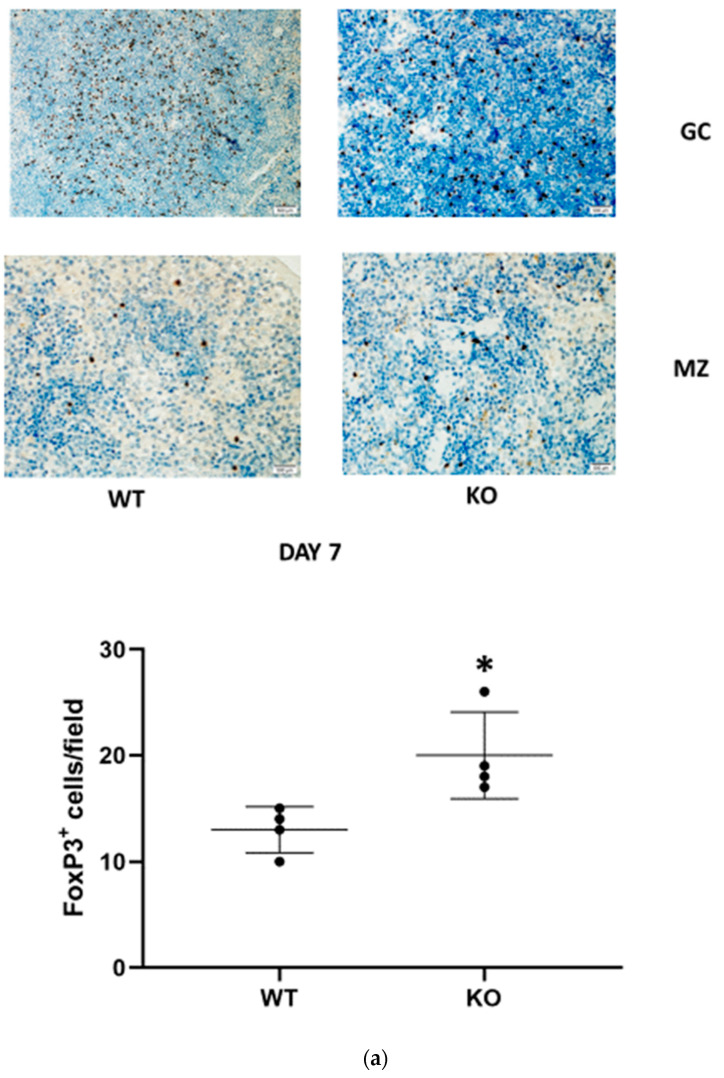
T regulatory cells (Tregs) are more abundant in CD24^−/−^ mice and attenuate renal damage 7 days following folic acid administration (FA-AKI). The presence of FoxP3+ T cells (Tregs), detected by immunohistochemistry, in the splenic germinal center (GC) and marginal zone (MZ) (**a**) and kidneys (**b**) is quantified as the ratio of FoxP3+/CD3+ cells per field, 4 fields/animal. A total of 4 mice per experimental group. Tissues were harvested from CD24^−/−^ and WT mice 7 days following folic acid administration. Original magnification ×400. Scale bars, 500 µm. Data are presented as the mean ± SD of 4 different experiments * *p* < 0.05 vs. the corresponding WT animals. (**c**). The decrease in serum creatinine (Δ serum creatinine) induced by the administration of anti CD25 antibodies to WT and KO mice 24 h prior to folic acid administration. The data are presented as the mean ± SD of eight mice per group. * *p* < 0.05 versus the corresponding untreated experimental group. Abbreviations: WT—wild type; KO—knockout.

**Figure 4 jpm-13-01134-f004:**
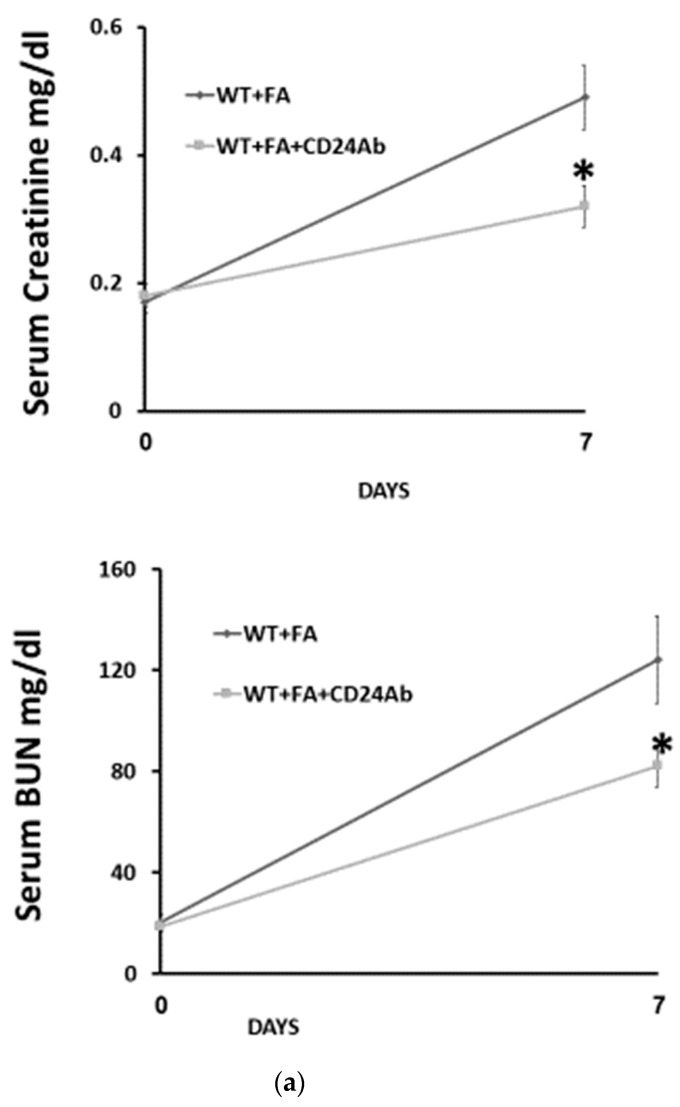
Anti-CD24 antibodies attenuate renal injury in folic acid-induced AKI. (**a**). Kidney function in response to administration of anti-CD24 or anti-isotype control antibodies to WT mice subjected to folic acid injury. Renal function was assessed by serum creatinine and BUN levels 7 days following the administration of folic acid. Renal failure was significantly attenuated in the anti CD24 antibody-treated FA-AKI mice compared with the corresponding untreated WT animals. The data are presented as the mean ± SD of eight mice per group. * *p* < 0.05 versus the corresponding untreated experimental group. (**b**). Decreased renal injury in WT mice exposed to FA and treated with anti-CD24 antibodies. Representative images of PAS staining and quantification on day 7 following folic acid administration. Mean ± SD of six mice per group. Original magnification ×400. Scale bars, 500 µm. * *p* < 0.05 versus the corresponding WT animal. Renal tubular and serum IL-33 were significantly higher in WT mice treated with anti-CD24 antibodies compared with untreated WT animals during FA-AKI. (**c**). Representative western blotting and densitometric analysis show upregulation of renal tubular IL-33 protein levels following the administration of folic acid. Data are presented as the mean ± SD of 3 different experiments * *p* < 0.05 vs. the corresponding WT animals. (**d**). Serum IL-33 levels 7 days following folic acid administration. Data are presented as the mean ± SD of 8 different experiments * *p* < 0.05 vs. the corresponding WT animals. Abbreviations: WT—wild type; Ab—antibodies; FA—folic acid.

**Figure 5 jpm-13-01134-f005:**
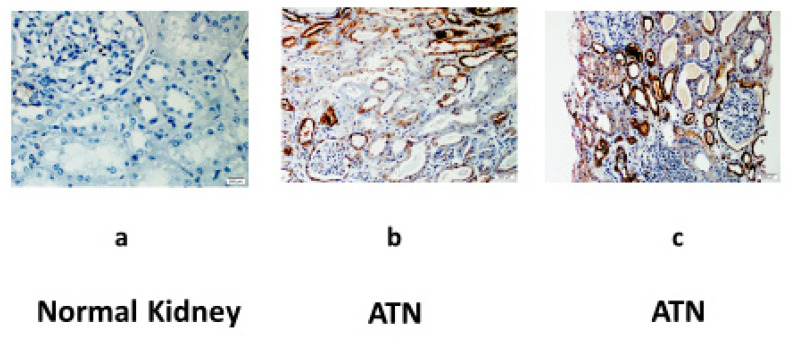
Renal CD24 expression detected by immunohistochemistry in patients with acute tubular necrosis (ATN). (**a**): Negative staining in normal renal tissue, which was removed along with renal cell carcinoma. Progressive positive staining for CD24 restricted to the distal tubular epithelial cells was observed in (**b**) a patient with ATN due to long-standing hypovolemia and (**c**) a patient with delayed graft function following a deceased donor kidney transplantation.

## Data Availability

All data are presented in the manuscript.

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
