# Peer review of "Renal Tubular CD24 Upregulation Aggravates Folic Acid Induced Acute Kidney Injury: A Possible Role for T Regulatory Cells Inhibition in Mice"

_jpm, 2023, doi:10.3390/jpm13071134_

Round 1
Reviewer 1 Report
I think it is a very well thought out study from a scientific point of view. However, especially the first paragraph of the introduction should be revised. Because information about acute kidney injury is out of context. The method and results sections give sufficient information. The discussion part is also satisfactory.
There are obvious spelling mistakes. I think the professional english edition can make the article more interesting and striking.
Author Response
The first paragraph in the introduction was revised and the "out of context" information regarding acute renal failure (page 3, lines 4—10 of the original version) was removed from the new version of the manuscript.
We have reviewed the manuscript to correct spelling mistakes. However, if the editorial board feels the additional corrections are required we agree to use the journal service.

Reviewer 2 Report
Moshe Shashar et.al, explored the function of CD24 in the folic acid induced acute kidney injury by CD24-/- mice, and reported FA- AKI CD24-/- mice exhibited an attenuated reduction in renal function and histological injury. Renal and systemic IL-33 upregulation were augmented. CD24-/- FA-AKI animals exhibited increased splenic margination and renal infiltration of regulatory T cells (Tregs) but exhibited increased expression of tubular pro-apoptotic and decreased anti-apoptotic proteins. Anti CD24 antibody administration to FA-AKI mice attenuated the decrease in renal function as well as histological injury. The study suggests the neutralization of CD24 may prove as a promising strategy for future therapies in AKI. However, there are substantial weakness in the manuscript that must be addressed before this could be published. Firstly, there are many mismatches between figure number and results. Besides, control groups were missing from a number of experiments.
Specific comments on the manuscript:
1. Figure 1: Western blotting result was missing from figure 1 and the figure numbers in the result section are mismatched with pictures. Control groups are needed in CD24 staining and PAS staining with higher quality and same scale bar is suggested.
2. Figure 2: The authors suggest that CD24 knockout increases IL33 expression level while CD24 mainly expresses in distal tubular cells. To clarify the source of IL33, co-staining of CD24 and IL33 is suggested. And a FA-AKI model with CD24 conditional knockout or bone marrow chimeric mice could rule out the role of immune cells.
3. Figure 3: Corresponding figure numbers were missing in result section. Is T regulatory cells infiltrating during recovery phase as the authors show that renal function is promoted only in day 7(Figure 1)? Therefore, data of FoxP3/CD3 ratio including day 1 and 3 are needed (Figure 3b). Both control and vehicle groups are needed in figure 3c. Why did renal function decline further more compared with WT group after anti-CD25 antibody administration?
4. Figure 4: What is the Y axis in figure 4c indicate? To elucidate leukocytes infiltration degrade, IHC staining of leukocytes is suggested.
Author Response
- The missing western blot in figure 1b was added to the new version of the manuscript and figure numbers now match the pictures.
- Figure 2b depicts the increase in renal tubular IL 33 during AKI which implies that IL 33 is generated by renal tubular cells and not inflammatory cells.
- TREGS infiltration during the earlier period of the study was negligible. This is why only data from day 7 are presented. We have modified the result section to clarify this point. (page 12 line 20).
- Figure 4c depicts renal tubular IL 33 content by western blotting and serum Il 33 levels.
Reviewer 3 Report
Moshe Shashar et. al have explored the role of CD24 in the pathogenesis of folic acid induced AKI (FA-AKI) in mice. Last figure with expression level of CD24 from three human kidney biopsies samples gives this manuscript higher dimension.
I recommend auto accept this manuscript after some minor corrections like
Please rewrite the abstract as an overall summary statement summery of the overall project rather than breaking down into background or methods and results
Space before “Diverse.. ” in page 4 line 2 (This sentence is itself too complex please rewrite as a simple sentence)
End parenthesis in page 6 line 3
Figure numbers in the manuscript is written as 1a, 1b etc with superscript, please re write them without superscription like 1a, 1b etc
Any comment on the mechanism behind CD24 inhibition with regulatory T cell development will be more helpful for the readers.
Thanking you
Author Response
- The abstract was modified as a summary statement
- The sentence in page 4 line 2 was simplified in the new version of the manuscript
- Corrected
- Corrected
- We have shown that during FA-AKI, CD 24 upregulation inhibits renal and systemic IL 33 (a known stimulator of TREGS). Therefore, we believe that the mechanism for TREG inhibition by CD24 involves IL33 (page 15 end of second paragraph).
Round 2
Reviewer 2 Report
The concerns have been addressed, but the conclusion of "CD24 aggravates the renal injury by inhibition Treg cells" should be further supported by additional experiments of dependence, so suggest to adjust the title of manuscript. And the details and description of revision suggest to be edit for easy reading.
-
Author Response
The concerns have been addressed, but the conclusion of "CD24 aggravates the renal injury by inhibition Treg cells" should be further supported by additional experiments of dependence, so suggest to adjust the title of manuscript. And the details and description of revision suggest to be edit for easy reading.
Response: The title of the manuscript was modified so that the role of TREGS is less affirmative.
